# Rethinking Visual Information Processing in Multimodal LLMs

## Abstract

Despite the remarkable success of the LLaVA architecture for vision-language tasks, its design inherently struggles to effectively integrate visual features due to the inherent mismatch between text and vision modalities. We tackle this issue from a novel perspective in which the LLM not only serves as a language model but also a powerful vision encoder. To this end, we present **LLaViT**–Large Language Models as extended Vision Transformers—which enables the LLM to simultaneously function as a vision encoder through three key modifications: (1) learning separate QKV projections for vision modality, (2) enabling bidirectional attention on visual tokens, and (3) incorporating both global and local visual representations. Through extensive controlled experiments on a wide range of LLMs, we demonstrate that LLaViT significantly outperforms the baseline LLaVA method on a multitude of benchmarks, even surpassing models with double its parameter count, establishing a more effective approach to vision-language modeling.

## 1 Introduction

Multimodal Large Language Models (MLLMs) Liu et al. (2023a; 2024); Dai et al. (2023); Tong et al. (2024a); Deitke et al. (2024) have emerged as a pivotal advancement in artificial intelligence. Leveraging the power and versatility of LLMs, these models enable us to tackle a wide range of tasks with a *single* model; tasks that previously required specialized models, such as image captioning and visual question answering, and even more traditional vision tasks such as object detection and image classification.

Among these MLLMs, LLaVA Liu et al. (2023a; 2024) stands out as a widely adopted framework with a simple architecture. It comprises of a pre-trained vision encoder Dosovitskiy et al. (2021); Radford et al. (2021), a pre-trained LLM Chiang et al. (2023), and a connector that projects visual features to the LLM's input dimensions. More recently, many works Deitke et al. (2024); Ranzinger et al. (2024); Chen et al. (2024a); Cha et al. (2024); Tong et al. (2024a); Chu et al. (2024); Chen et al. (2024b); Wang et al. (2023a) have proposed various improvements to the LLaVA framework, focusing on developing stronger vision encoders Ranzinger et al. (2024); Chen et al. (2024a); Wang et al. (2024), designing sophisticated connector architectures Cha et al. (2024); Tong et al. (2024a); Chu et al. (2024), or compiling higher quality training datasets Deitke et al. (2024); Chen et al. (2024b); Wang et al. (2023a).

In our work, we explore the LLaVA framework from a novel perspective. A conventional of understanding of LLaVA-like architectures suggests that, through its pre-training stage, the visual features become aligned with the LLM's input space, thereby allowing the LLM to process visual tokens similarly to text tokens. From this point of view, there is a clear distinction between the roles of the vision encoder and the LLM. However, our investigations show that the visual tokens at the input layer of the LLM are *not* well aligned with the LLM's input space; rather, the LLM itself gradually *translates* visual representations to text representations, progressively aligning the two modalities in its transformer layers. Moreover, we find that attention updates to the visual tokens *within* the LLM have a profound impact on the LLM's ability to process visual information. Based on these insights, we propose a new perspective of MLLMs that has not been extensively explored before. Instead of viewing the vision encoder and LLM as two separate components with distinct roles, we consider the vision encoder as extending into the LLM itself. In other words, the LLM

serves not only as a language processor that understands prompts and generates answers, but also as an integral part of the visual feature processing pipeline.

This new perspective motivates **LLaViT**, **L**arge **La**nguage Models as extended **Vi**sion **T**ransformers, which consists of three simple yet effective approaches to *transform* the LLM to additionally serve as a powerful vision encoder: (i) learning separate QKV projections for visual tokens, (ii) enabling bidirectional attention on visual tokens, and (iii) incorporating both local and global features from the (original) vision encoder. When integrating LLaViT to the LLaVA framework with various LLMs, we observe substantial performance gains across a wide range of MLLM benchmarks. Notably, on several key vision-oriented benchmarks, our 3B LLaViT model not only *outperforms* the 7B LLaVA-1.5 Liu et al. (2024) model but also achieves performance comparable to the 14B LLaVA-1.5 model. This demonstrates the effectiveness of our method and establishes a promising new direction for MLLM architecture design.

## 2 BACKGROUND AND MOTIVATION

To set the stage, we first provide an overview of the LLaVA Liu et al. (2023a; 2024) model and establish key notations. We then discuss two preliminary investigations that motivate our work.

### 2.1 REVIEW OF LLAVA

LLaVA tackles vision-language tasks by treating visual information as specialized input embeddings for a pre-trained LLM. Consider an LLM $h_\theta$ with $L$ transformer Vaswani et al. (2017) layers, parameterized by $\theta$. We represent the general case of multimodal inputs to the LLM's $\ell$-th layer as a combination of three distinct sequences: (1) $m$ text tokens for the system prompt, $\mathbf{t}_{\text{sys}}^\ell = (t_1^\ell, t_2^\ell, \ldots, t_m^\ell)$, (2) $n$ visual tokens for the visual information, $\mathbf{v}^\ell = (v_1^\ell, v_2^\ell, \ldots, v_n^\ell)$, and (3) $o$ text tokens for the user prompt, $\mathbf{t}_{\text{usr}}^\ell = (t_{m+1}^\ell, t_{m+2}^\ell, \ldots, t_{m+o}^\ell)$. At any given layer $\ell$, $\mathbf{t}_{\text{sys}}^\ell$, $\mathbf{v}^\ell$, and $\mathbf{t}_{\text{usr}}^\ell$ are processed as a single $N = m + n + o$ length sequence,

$$\mathbf{x}^\ell = (x_1^\ell, x_2^\ell, \ldots, x_N^\ell) = (\mathbf{t}_{\text{sys}}^\ell, \mathbf{v}^\ell, \mathbf{t}_{\text{usr}}^\ell), \tag{1}$$

where $x_i^\ell$ represents the $i$-th input token in the $\ell$-th layer[1], and the set of indices corresponding to the visual tokens can be defined as

$$\mathcal{I}_v = \{m+1, m+2, \ldots, m+n\}. \tag{2}$$

Given an input image $I$, we extract the visual patch features using a pre-trained vision encoder, $g$, then project them to the LLM's embedding space with an MLP projection, $f_\phi : \mathbb{R}^{d_V} \to \mathbb{R}^{d_L}$, parameterized by $\phi$, where $d_V$ and $d_L$ represent the feature dimensions of the vision encoder and LLM, respectively:

$$\mathbf{v}^1 = f_\phi(g(I)) = (v_1^1, v_2^1, \ldots, v_n^1). \tag{3}$$

To train the model, LLaVA employs a two-stage training pipeline. The first stage, referred to as *pre-training*, aims to align the visual token embeddings $\mathbf{v}$ with the LLM's input embedding space using image-text pairs. Here, both the vision encoder $g$ and the LLM $h_\theta$ kept frozen, while the parameters of the MLP projection $\phi$ are trained. The second stage, referred to as *fine-tuning* or *instruction tuning*, uses image-question-answer triplets to fine-tune the parameters of both the MLP projection and the LLM, $\{\phi, \theta\}$.

### 2.2 MLLMS TRANSLATE VISUAL TOKENS TO TEXT

The LLM $h_\theta$ processes text data by first tokenizing the input text into a sequence of tokens, then embedding the tokenizations with a set of word embeddings, $\mathcal{W} = \{w_1, w_2, \ldots, w_M\}$. While the LLM's input is confined to the discrete space defined by $\mathcal{W}$, the input layer's visual tokens, $\mathbf{v}^1$, are not constrained to a discrete space. However, given that the pre-training stage is dedicated to align the visual embeddings with text, one may naturally expect $\mathbf{v}^1$ to be aligned with $\mathcal{W}$.

---

[1]Without loss of generality, we omit the layer index $\ell$ when the specific layer is irrelevant.

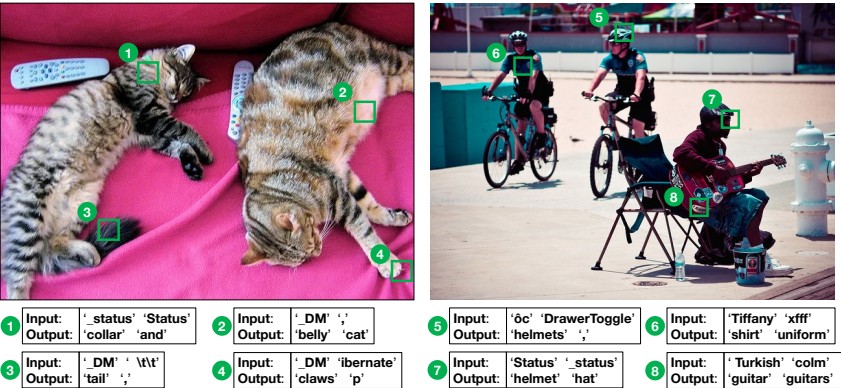

Figure 1: Visualizing how the LLM interprets visual tokens at the input and output layers of the LLM. Input layer word representations are selected using Eq. (4). Output layer word representations are selected based on the LLM's final logits of the visual token. For better interpretability, we manually select 2 of the top-3 word representations for each of the selected visual tokens.

To further investigate this, we first train a LLaVA model using a `Qwen2.5-3B` Yang et al. (2024) as the LLM. Then, for an input layer visual token $v_i^1$, we can compute the cosine similarity between $v_i^1$ and a word embedding $w \in \mathcal{W}$ as:

$$\Omega_i(w) = \frac{v_i^1 \cdot w}{\|v_i^1\|\|w\|}. \tag{4}$$

By computing the $\Omega_i(w)$ for all $w \in \mathcal{W}$, we can extract the top-$k$ similar words for each visual token $v_i^1$. We visualize these word representations for select visual tokens in Figure 1 and observe that the closest word representations are vastly unrelated to the corresponding image patch, often matching to unnatural strings such as "_DM", "_status" or "\t\t". Furthermore, we notice that top cosine similarities are low ($\Omega_i(w) \simeq 0.1$ for most patches[2]), indicating a significant modality gap between visual and text embeddings in the input space.

We extend our investigation to the LLM's output of the visual tokens, $\mathbf{z} = (z_1, z_2, \ldots, z_n) = h_\theta(\mathbf{v}^1)$, where $z_i$ denotes the output logit vector for $v_i$. Following a similar approach to the logit lens nostalgebraist (2020), we employ $z_i$ to extract the top-$k$ word representations from $v_i$ and visualize the results in Figure 1. Surprisingly, at the LLM's output, we observe that the word representations for visual embeddings are often relevant to the corresponding image patch, *i.e.*, the LLM can *translate* visual tokens into text to some extent. For example, the LLM correctly predicts the "tail" and "belly" of the cat (Figure 1 left), as well as the "helmets", "guitar", and "uniform" (Figure 1 right).

What's particularly intriguing is that such translation of visual tokens is never explicitly supervised; rather, it *emerges* from supervision on text tokens. Moreover, as shown in Figure 1, the visual tokens are not well aligned with the text tokens in the input layer, suggesting that the LLM actively translates/aligns visual representations to text representations because it is necessary to interpret visual information. Thus, we posit that the quality of visual token transformations within the LLM will have a profound effect on the MLLM's overall capabilities.

## 2.3 IMPORTANCE OF VISUAL ATTENTION IN MLLMs

We continue our investigation by examining the importance of attention *between* visual tokens within the LLM. While the cross-attention between vision and text modalities is essential for information to flow from vision to text, it remains unclear whether visual tokens need to attend to each other within the LLM, especially considering that they have already undergone attention updates in the vision encoder. To facilitate our investigation, we conduct an ablation study by training a LLaVA model *without* visual attention in all layers of the LLM. More specifically, we modify the attention layer of

---

[2]We provide a detailed version of Figure 1 with the cosine similarities in the Appendix.

Table 1: Comparison of LLaVA-1.5 (baseline) with and without visual attention, as described in Eq. (5). Experiments are conducted with `Qwen2.5-3B` Yang et al. (2024) and `7B`. A breakdown of the benchmarks in each category can be found in Section 4.1, and the full table is provided in the Appendix.

| Method | Vision Centric | OCR & Chart | Knowledge | General |
|---|---|---|---|---|
| | `Qwen2.5-3B` | | | |
| Baseline | 39.3 | 27.4 | 67.5 | 65.9 |
| No Visual Attention | 24.9 (-14.4) | 10.7 (-16.7) | 64.6 (-2.9) | 50.2 (-15.7) |
| | `Qwen2.5-7B` | | | |
| Baseline | 45.0 | 31.8 | 71.7 | 68.5 |
| No Visual Attention | 40.8 (-4.2) | 27.6 (-4.2) | 69.8 (-1.9) | 67.0 (-1.5) |

the LLM such that:

$$x_i' = \begin{cases} x_i & \text{if } i \in \mathcal{I}_v \\ \text{CausalAttn}(x_i) + x_i & \text{otherwise,} \end{cases} \quad (5)$$

where $x_i$ and $x_i'$ each represent an input and output token of the attention layer respectively, and CausalAttn($x_i$) represents the causal attention layer in which $x_i$ serves as the query vector. Note that Eq. (5) only disables attention updates when $x_i$ is a visual token, meaning that a text token, $x_j$ where $j > i$ and $j \notin \mathcal{I}_v$, can still attend to visual tokens. In addition, despite removing the attention updates from visual tokens, we do *not* restrict the forward pass to the MLP of each transformer layer.

We evaluate these models on 17 benchmarks (grouped into 4 distinct categories; refer to Section 4.1 for more details) and compare with the baseline LLaVA model in Table 1. Overall, the model *without* visual attention exhibits significantly degraded performance across all categories on both Qwen2.5-3B and Qwen2.5-7B. This degradation is particularly pronounced in the Vision Centric and OCR & Chart categories, which rely more heavily on visual information than the Knowledge and General categories. Thus, our investigation demonstrates that attention between visual tokens *within* the LLM do indeed play a critical role in the MLLM, further corroborating the argument that higher quality of visual token transformations within the LLM are crucial for strong performance.

## 3 LLAViT: Extending the Vision Transformer to the LLM

We now present **LLaViT**, which consists of three key enhancements that allow the LLM to serve as an extended Vision Transformer. Motivated by our investigations from Sections 2.2 and 2.3, in Sections 3.1 and 3.2 we detail the enhancements that focus on improving the visual information processing *within* the LLM—a direction that has not been extensively explored in previous works. In Section 3.3, we present a simple yet effective method to enhance the quality of input visual tokens without sacrificing efficiency.

### 3.1 Learning Separate QKV Projections for Visual Tokens

As discussed in Section 2.2, there is a clear misalignment between input text and visual tokens. This misalignment can lead substantial challenges, especially in the LLM's attention layers, where visual tokens undergo attention updates based on parameters that were trained specifically for the text modality. To facilitate better visual representation learning in the LLM, we propose a modality-specific attention mechanism by separating the Query, Key, and Value (QKV) projection parameters for text and visual tokens.

Let $\{W_Q^{\text{text}}, W_K^{\text{text}}, W_V^{\text{text}}\}$ represent the attention layer's QKV projection parameters, trained extensively on text data. We copy these parameters into new *visual* QKV parameters $\{W_Q^{\text{vis}}, W_K^{\text{vis}}, W_V^{\text{vis}}\}$, which are used exclusively to project visual tokens. Formally, given an arbitrary token $x_i$, the corresponding query vector $\mathbf{q}_i$ is computed as:

$$\mathbf{q}_i = \begin{cases} W_Q^{\text{vis}} x_i & \text{if } i \in \mathcal{I}_v \\ W_Q^{\text{text}} x_i & \text{otherwise,} \end{cases} \quad (6)$$

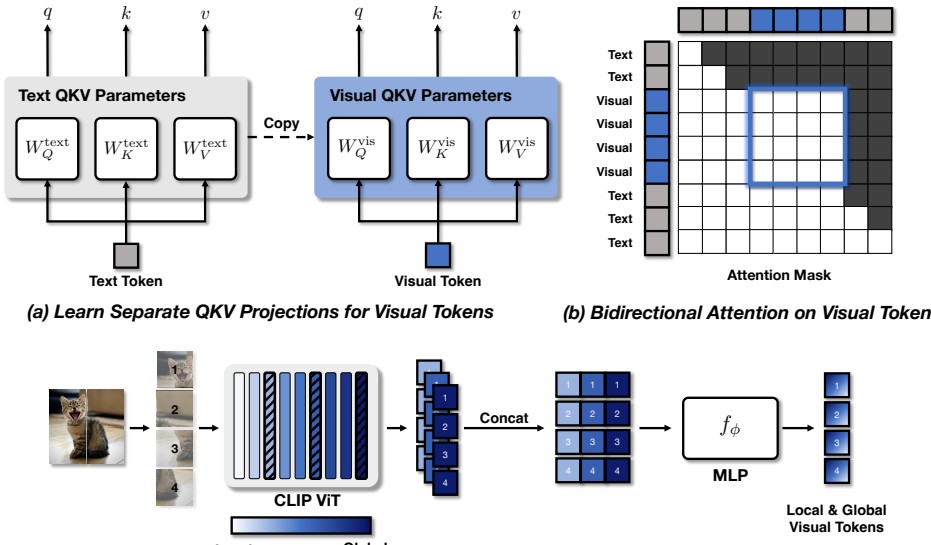

Figure 2: Overview of LLaViT, which transforms the LLM to act as an extended vision encoder. (a) We learn separate QKV projection parameters for visual tokens, initialized with the weights of the LLM's QKV parameters. (b) While the LLM employs causal attention on *all* tokens, we enable bidirectional attention on the visual tokens. (c) We incorporate both local and global features in the visual tokens by extracting patch features from multiple layers of the CLIP ViT model.

and similarly for the key and value vectors, $\mathbf{k}_i$ and $\mathbf{v}_i$. We apply these separate QKV projections on all layers of the LLM, and unlike the original QKV parameters, we tune the visual QKV parameters $\{W_Q^{\text{vis}}, W_K^{\text{vis}}, W_V^{\text{vis}}\}$ during the pre-training stage. This enables the LLM to leverage image-caption data to not only learn stronger visual representations, but also to better align visual representations to the text representations.

This separation also addresses another crucial challenge from an optimization perspective. In the fine-tuning stage, the LLM must simultaneously adapt to visual tokens while maintaining language understanding capabilities, creating conflicts in model optimization. This relates to the stability-plasticity dilemma Kim & Han (2023): On one hand, the LLM may prioritize stability, making only minor adaptations (*e.g.* following instructions or formatting responses), thereby failing to process visual information effectively. On the other hand, prioritizing plasticity to enhance visual information risks degrading the LLM's fundamental language capabilities. Our approach mitigates this issue by compartmentalizing the visual adaptation process, allowing a dedicated optimization of visual representations whilst preserving the LLM's core knowledge.

## 3.2 BIDIRECTIONAL ATTENTION

In Section 2.3, we identified that attention updates to visual tokens within the LLM play a crucial role in the performance of the MLLM. However, LLMs employ causal attention, which artificially restricts attention among visual tokens, only allowing "later" visual tokens to attend to "earlier" ones but not vice versa. While such causality is valid for text generation, it leads to a severe imbalance of attention updates on visual tokens that have no inherent temporal ordering. To mitigate this issue, we enable bidirectional attention on the visual tokens.

Given a query vector $\mathbf{q}_i = W_q x_i$ and a key vector $\mathbf{k}_j = W_k x_j$, the softmax attention score $p_{ij}$ in causal attention can be formulated as:

$$p_{ij} = \frac{\exp(s_{ij})}{\sum_{k=1}^{N} \exp(s_{ik})}, \quad s_{ij} = \begin{cases} (\mathbf{q}_i \cdot \mathbf{k}_j)/\sqrt{d_L} & \text{if } j \leq i \\ -\infty & \text{otherwise,} \end{cases} \quad (7)$$

where the $-\infty$ ensures that attention is masked when $j > i$. To enable bidirectional attention on the visual tokens, we modify the case function of Eq. (7) such that $s_{ij}$ is now defined as:

$$s_{ij} = \begin{cases} (\mathbf{q}_i \cdot \mathbf{k}_j)/\sqrt{d_L} & \text{if } j \leq i \textbf{ or } i, j \in \mathcal{I}_v \\ -\infty & \text{otherwise,} \end{cases} \tag{8}$$

where now the masking does not apply to visual tokens, *i.e.*, when $i, j \in \mathcal{I}_v$. We illustrate the attention mask defined by Eq. (8) in Figure 2(b).

### 3.3 LOCAL AND GLOBAL FEATURES

The visual tokens of Eq. (3) are extracted from the penultimate transformer layer of the CLIP Radford et al. (2021) image encoder, which has been trained to prioritize global semantic alignment between images and text Monsefi et al. (2024) and often struggles to capture fine-grained details Ghiasi et al. (2022). To compensate for this information loss, we extract visual features from 3 different depths of the CLIP vision encoder, creating a representation that combines both high-level semantic information and lower-level details. Then, we concatenate the visual features along the feature dimension and project them to the LLM's input dimensions with the MLP projection $f_\phi : \mathbb{R}^{3d_V} \to \mathbb{R}^{d_L}$. We illustrate this process in Figure 2(c).

Our approach increases the visual information density per token, providing the LLM with a spectrum of visual information ranging from local details to global semantic context. Despite this, we still maintain the same computational efficiency as before since we extract multiple features from the same vision encoder within a single forward pass. Furthermore, by concatenating features along the feature dimension rather than the token dimension, we avoid increasing the number of visual tokens input to the LLM, which would otherwise substantially increase computational costs.

## 4 EXPERIMENTS

To evaluate the effectiveness of LLaViT, we trained multiple models using various base LLMs and conducted evaluations on a wide variety of MLLM benchmarks.

### 4.1 EXPERIMENTAL SETTING

**Models.** For the vision encoder, we follow LLaVA-1.5 and use `OpenAI CLIP ViT-L/14` Radford et al. (2021) with 336px resolution. To demonstrate the effectiveness of LLaViT on a wide range of LLMs, we mainly experiment with the instruction tuned versions of `Qwen2.5` Yang et al. (2024), using various-sized LLMs including `1.5B`, `3B`, `7B`, and `14B` parameter models. We also experiment using `Phi-3.5-mini-instruct` Abdin et al. (2024) and present the results in the Appendix.

**Training data.** For the pre-training data, we use the PixMo-Cap Deitke et al. (2024) dataset. While the PixMo-Cap originally contains 712k distinct images, we train on a 622k subset after filtering broken URLs and faulty image files. Compared to the original LLaVA pre-training dataset, PixMo-Cap provides higher-quality human annotated image-caption pairs with fine-grained and dense captions. For instruction tuning, we use the LLaVA-1.5 instruction tuning dataset Liu et al. (2024) with 665k samples. To ensure fair comparison across the board, both the baseline and LLaViT models are pre-trained on PixMo-Cap and fine-tuned on the LLaVA-1.5.

**Evaluation benchmarks.** We evaluate on a large suite of 17 MLLM benchmarks using the `lmms-eval` library Zhang et al. (2024). For better interpretability of our experimental results, we group the 17 benchmarks based on the categorizations defined in Tong et al. (2024a):

**Vision Centric (2):** RealWorldQA xAI (2024), MMVP Tong et al. (2024b).
**OCR & Chart (5):** ChartQA Masry et al. (2022), DocVQA Mathew et al. (2021), InfoVQA Mathew et al. (2022), OCRBench Liu et al. (2023d), TextVQA Singh et al. (2019).
**Knowledge (2):** Science-QA Lu et al. (2022), AI2D Hiippala et al. (2021).
**General (8):** GQA Hudson & Manning (2019), MMBench-EN Liu et al. (2023c), MMBench-CN Liu et al. (2023c), MME (perception) Fu et al. (2023), POPE Li et al. (2023), VizWiz Gurari et al. (2018), MMStar Chen et al. (2024c), VQAv2 Goyal et al. (2017).

Table 2: Evaluation results on 17 MLLM benchmarks using `Qwen2.5-1.5B`, `3B`, `7B` and `14B` as the base LLM. "Baseline" refers to LLaVA-1.5 Liu et al. (2024) trained with the respective LLMs, and the *Any-Res* setting is denoted by the `-HD` suffix. We present evaluation results on individual benchmarks, as well as the averages of each category, colored in blue.

| Method | Vision Centric | | | OCR & Chart | | | | | | Knowledge | | | General | | | | | | | | |
| | RealWorldQA | MMVP | Avg | ChartQA | DocVQA | InfoVQA | OCRBench | TextVQA | Avg | SciQA | AI2D | Avg | GQA | MMBench-EN | MMBench-CN | MME-P | POPE | VizWiz | MMStar | VQAv2 | Avg |
|---|---|---|---|---|---|---|---|---|---|---|---|---|---|---|---|---|---|---|---|---|---|
| | | | | | | | | | `Qwen2.5-1.5B` | | | | | | | | | | | | |
| Baseline | 52.4 | 21.3 | 36.9 | 16.8 | 22.1 | 19.9 | 31.8 | 41.2 | 26.3 | 70.6 | 58.5 | 64.6 | 60.2 | 65.7 | 60.7 | 1395.3 | **85.9** | 48.5 | 39.4 | 74.4 | 63.1 |
| LLaViT | 53.7 | 29.3 | **41.5** | 22.2 | 26.6 | 21.6 | 34.4 | 46.2 | **30.2** | 70.9 | 61.5 | **66.2** | 61.2 | 67.5 | 61.3 | 1421.5 | 85.8 | 50.5 | 41.8 | 76.1 | **64.4** |
| | | | | | | | | | `Qwen2.5-3B` | | | | | | | | | | | | |
| Baseline | 54.5 | 24.0 | 39.3 | 17.6 | 22.8 | 22.0 | 32.3 | 42.4 | 27.4 | **73.4** | 61.6 | 67.5 | 61.3 | 71.0 | 66.6 | 1451.7 | 85.8 | 49.7 | 44.0 | 75.9 | 65.9 |
| LLaViT | 56.5 | 38.7 | **47.6** | 23.1 | 28.7 | 23.9 | 37.1 | 48.5 | **32.2** | 72.8 | 63.8 | **68.3** | 62.5 | 72.2 | 68.0 | 1453.3 | 86.2 | 52.0 | 46.4 | 77.6 | **67.2** |
| | | | | | | | | | `Qwen2.5-7B` | | | | | | | | | | | | |
| Baseline | 58.7 | 31.3 | 45.0 | 23.0 | 27.0 | 24.4 | 36.3 | 48.1 | 31.8 | 75.7 | 67.6 | 71.7 | 63.2 | 71.1 | 68.0 | 1506.6 | **87.1** | 58.7 | 46.2 | 78.4 | 68.5 |
| LLaViT | 59.7 | 41.7 | **50.7** | 27.1 | 31.9 | 27.1 | 40.4 | 52.0 | **35.7** | 76.7 | 68.7 | **72.7** | 63.9 | 74.7 | 73.2 | 1591.5 | 86.3 | 62.6 | 49.4 | 79.6 | **71.2** |
| | | | | | | | | | `Qwen2.5-14B` | | | | | | | | | | | | |
| Baseline | 59.9 | 32.7 | 46.3 | 23.4 | 27.3 | 27.1 | 35.0 | 49.1 | 32.4 | 77.7 | 71.0 | 74.4 | 64.3 | 76.7 | 75.0 | 1594.8 | 86.5 | 64.4 | 46.4 | 79.2 | 71.5 |
| LLaViT | 61.6 | 40.7 | **51.2** | 31.7 | 34.3 | 30.6 | 39.9 | 54.0 | **38.1** | 80.0 | 73.9 | **77.0** | 65.1 | 77.2 | 76.2 | 1670.6 | 87.0 | 66.8 | 51.1 | 80.5 | **73.4** |
| | | | | | | | | | `Qwen2.5-3B-HD` | | | | | | | | | | | | |
| Baseline | 57.0 | 33.0 | 45.0 | 25.4 | 53.1 | 32.5 | 43.1 | 60.0 | 42.8 | **73.1** | 62.4 | 67.8 | 63.1 | 70.2 | 65.7 | 1445.9 | 87.0 | 51.4 | 46.4 | 78.7 | 66.8 |
| LLaViT | 57.9 | 40.0 | **49.0** | 31.4 | 59.4 | 35.4 | 48.8 | 65.4 | **48.1** | 73.0 | 64.4 | **68.7** | 64.1 | 71.9 | 69.5 | 1488.7 | 87.6 | 54.4 | 48.0 | 80.0 | **68.7** |
| | | | | | | | | | `Qwen2.5-7B-HD` | | | | | | | | | | | | |
| Baseline | 63.9 | 30.7 | 47.3 | 30.9 | 57.6 | 35.8 | 49.4 | 64.4 | 47.6 | 75.9 | 67.4 | 71.7 | 64.4 | 75.1 | 70.8 | 1575.4 | **87.9** | 57.7 | 47.8 | 80.9 | 70.4 |
| LLaViT | 64.6 | 41.3 | **53.0** | 40.3 | 63.9 | 39.4 | 54.1 | 67.8 | **53.1** | 77.1 | 69.6 | **73.4** | 65.5 | 76.3 | 73.5 | 1625.0 | 87.9 | 55.2 | 48.6 | 81.7 | **71.2** |
| | | | | | | | | | `Qwen2.5-14B-HD` | | | | | | | | | | | | |
| Baseline | 65.8 | 37.3 | 51.5 | 39.0 | 55.8 | 37.3 | 45.7 | 64.5 | 48.5 | 78.5 | 71.3 | 74.9 | 66.2 | 78.0 | 76.1 | 1638.9 | **87.5** | 62.8 | 49.0 | 81.6 | 72.9 |
| LLaViT | 66.4 | 45.3 | **55.9** | 46.5 | 67.6 | 44.4 | 56.5 | 70.0 | **57.0** | 79.5 | 74.7 | **77.1** | 66.3 | 79.4 | 78.7 | 1683.0 | 87.3 | 64.8 | 52.5 | 82.7 | **74.5** |

**Implementation details.** For the visual QKV parameters, we use a learning rate of 2e-4 with a cosine decay schedule Loshchilov & Hutter (2017), and for local/global visual features, we extract patch features from the 5th, 15th, and 23rd layer of `CLIP ViT-L/14`. We follow LLaVA-1.5 for other hyperparameters such as pre-train/fine-tune learning rate, optimizer choice, number of epochs, and train all models with a fixed global batch size of 256 and 128 for pre-training and fine-tuning, respectively. Furthermore, we utilize DeepSpeed Rajbhandari et al. (2020) ZeRO-2 for pre-training and ZeRO-3 for fine-tuning, and use FlashAttention2 Dao et al. (2022); Dao (2024) as the attention implementation. We train and evaluate our models on two input resolution settings: (i) the *Standard-Res* setting, resizing all images to a single $336 \times 336$px image, and (ii) the *Any-Res* (HD) setting, where we additionally split images into smaller, non-overlapping $336 \times 336$px images that are processed individually by the vision encoder. Compared to the Standard-Res setting, which uses 576 visual tokens, the Any-Res setting uses upto 2880 visual tokens.

## 4.2 RESULTS ON STANDARD-RES

The upper section of Table 2 presents and compares the evaluation results of LLaViT against LLaVA-1.5 (baseline) the standard-res setting.

**Vision Centric and OCR&Chart.** LLaViT excels in Vision Centric and OCR & Chart tasks. In the Vision Centric category, LLaViT improves over the baseline by 4.6pp, 8.3pp, 5.7pp, and 4.9pp for the `1.5B`, `3B`, `7B`, and `14B` LLMs, respectively. Similarly in the OCR & Chart category, LLaViT outperforms the baseline by 3.9pp, 4.8pp, 3.9pp, and 5.7pp. LLaViT also consistently outperforms *all* individual benchmarks of these two categories. Of particular interest is the MMVP Tong et al. (2024b) benchmark, which is known to be challenging even for production-grade MLLMs such as GPT-4V OpenAI (2023) and Gemini Google (2023), which are reported to have accuracy of 38.7% and 40.7%, respectively. Both our `7B` and `14B` models match or outperform these two production models, and across the board we observe improvements between 8.0pp and 14.7pp over the baseline. Another remarkable observations is that our approach, even when using a smaller LLM, outperforms baseline models that use LLMs with double the size. For example, our `7B` model outperforms the

Table 3: Results of ablation experiments with `Qwen2.5-3B` and `7B` as the base LLM. "Sep. QKV", "BiAttn", and "Local/Global" refer to the three components to LLaViT discussed in Sections 3.1, 3.2, and 3.3, respectively.

| Row | Method | Vision Centric | | | OCR & Chart | | | | | | Knowledge | | | General | | | | | | | | |
| | | RealWorldQA | MMVP | Avg | ChartQA | DocVQA | InfoVQA | OCRBench | TextVQA | Avg | SciQA | AI2D | Avg | GQA | MMBench-EN | MMBench-CN | MME-P | POPE | VizWiz | MMStar | VQAv2 | Avg |
|---|---|---|---|---|---|---|---|---|---|---|---|---|---|---|---|---|---|---|---|---|---|---|
| | | | | | | | | | | Qwen2.5-3B | | | | | | | | | | | | |
| ❶ | Baseline | 54.5 | 24.0 | 39.3 | 17.6 | 22.8 | 22.0 | 32.3 | 42.4 | 27.4 | 73.4 | 61.6 | 67.5 | 61.3 | 71.0 | 66.6 | 1451.7 | 85.8 | 49.7 | 44.0 | 75.9 | 65.9 |
| ❷ | ❶ + Sep. QKV | 55.3 | 34.0 | 44.7 | 20.2 | 26.8 | 23.5 | 35.6 | 45.9 | 30.4 | **74.5** | 61.9 | 68.2 | 62.2 | 70.3 | 68.4 | 1441.2 | **87.1** | 49.6 | 44.8 | 76.8 | 66.4 |
| ❸ | ❷ + BiAttn | 54.1 | 36.6 | 45.4 | 22.8 | 27.7 | **23.9** | 37.3 | 47.5 | 31.8 | 72.9 | **63.8** | 68.4 | 62.1 | 70.8 | 68.1 | 1455.1 | 86.0 | 50.7 | **46.4** | 77.2 | 66.8 |
| ❹ | ❷ + Local/Global | 55.0 | **40.0** | 47.5 | **23.3** | 27.5 | 23.5 | 37.1 | 47.5 | 31.8 | 74.1 | 62.9 | **68.5** | **62.5** | 71.4 | **69.2** | **1462.8** | 86.4 | 50.7 | 46.1 | 77.3 | 67.1 |
| ❺ | LLaViT | **56.5** | 38.7 | **47.6** | 23.1 | **28.7** | **23.9** | 37.1 | **48.5** | **32.2** | 72.8 | **63.8** | 68.3 | **62.5** | **72.2** | 68.0 | 1453.3 | 86.2 | **52.0** | **46.4** | **77.6** | **67.2** |
| | | | | | | | | | | Qwen2.5-7B | | | | | | | | | | | | |
| ❶ | Baseline | 58.7 | 31.3 | 45.0 | 23.0 | 27.0 | 24.4 | 36.3 | 48.1 | 31.8 | 75.7 | 67.6 | 71.7 | 63.2 | 71.1 | 68.0 | 1506.6 | **87.1** | 58.7 | 46.2 | 78.4 | 68.5 |
| ❷ | ❶ + Sep. QKV | 59.7 | 34.7 | 47.2 | 23.5 | 28.5 | 25.5 | 37.7 | 49.4 | 32.9 | 77.6 | 67.7 | 72.6 | 63.2 | 71.7 | 69.6 | 1559.7 | 86.6 | 60.0 | 46.8 | 78.8 | 69.3 |
| ❸ | ❷ + BiAttn | **60.9** | 34.7 | 47.8 | 24.8 | 30.7 | 26.7 | 38.9 | 50.6 | 34.3 | 78.2 | 68.0 | 73.1 | 63.4 | 73.3 | 70.6 | **1606.3** | 86.8 | 59.9 | 46.7 | 79.1 | 70.0 |
| ❹ | ❷ + Local/Global | **60.9** | 36.7 | 48.8 | 26.1 | 30.1 | 26.3 | 40.1 | 50.7 | 34.7 | **78.5** | **69.0** | **73.8** | **64.0** | 73.0 | 71.1 | 1567.9 | **87.1** | 58.9 | 48.7 | 79.3 | 70.0 |
| ❺ | LLaViT | 59.7 | **41.7** | **50.7** | **27.1** | **31.9** | **27.1** | 40.4 | **52.0** | **35.7** | 76.7 | 68.7 | 72.7 | 63.9 | **74.7** | **73.2** | 1591.5 | 86.3 | **62.6** | 49.4 | 79.6 | **71.2** |

`14B` baseline by 4.4pp on Vision Centric (50.7% vs. 46.3%), and 3.3pp on OCR & Chart (35.7% vs. 32.4%). In addition, our `3B` model (47.6% for Vision Centric and 32.2% for OCR & Chart) performs on par with the `14B` baseline (46.3% for Vision Centric and 32.4% for OCR & Chart) despite having less than a quarter of the parameters[3]. This clearly demonstrates the effectiveness of LLaViT and highlights the importance of enhancing visual information processing within the LLM.

**Knowledge and General.** LLaViT also improves over the baseline on the Knowledge and General categories, showing gains of 1.6pp, 0.8pp, 1.0pp, and 2.6pp on Knowledge and gains of 1.3pp, 1.3pp, 2.7pp, and 1.9pp on the General category for the `1.5B`, `3B`, `7B`, and `14B` LLMs. While these improvements remain consistent across different model sizes and individual benchmarks, the gains in the Knowledge category are relatively modest compared to other areas. In fact, this aligns with our expectations since the Knowledge benchmarks are highly dependent on the LLM's inherent knowledge, whereas as our approach primarily focuses on enhancing the LLM's ability to understand visual information.

### 4.3 RESULTS ON ANY-RES

We present evaluation results of models trained under the Any-Res setting in the bottom section of Table 2, denoted by the `-HD` suffix. Compared to the Standard-Res setting, we observe that the *baseline* performance is particularly affected in the Vision Centric and OCR & Chart categories, while the Knowledge and General categories show less significant changes. This aligns with our expectations, as Vision-Centric and OCR & Chart categories heavily depend on visual information, making increased input granularity especially beneficial for performance in these areas.

When comparing between the baseline and LLaViT, we observe similar trends to those of the standard-res setting. LLaViT exhibits significant gains of 4.0pp, 5.7pp, and 4.4pp on Vision Centric for the `3B`, `7B`, and `14B` `Qwen2.5` LLMs. Moreover, on OCR & Chart LLaViT improves over the baseline by 5.3pp, 5.5pp, and 8.5pp. Finally, much like the standard-res models, the smaller variant of LLaViT outperforms the larger baseline models on these two categories, *i.e.* the `3B-HD` and `7B-HD` LLaViT outperform the `7B-HD` and `14B-HD` baselines, respectively.

### 4.4 ABLATIONS

We conducted ablation experiments on the three components to highlight their contributions. To demonstrate consistent trends across different sizes of LLMs, we conducted these experiments using both `Qwen2.5-3B` and `7B` as the base LLM, and present these results in Table 3. We notice progressive performance improvements as each method is applied to the baseline. First, we observe significant improvements in both models across all benchmark categories just by adding separate

---

[3]Despite learning separate QKV projections for visual tokens, the parameter increase is limited to 5%∼12%. We provide a thorough analysis of the memory and computational overhead of LLaViT in Appendix A.

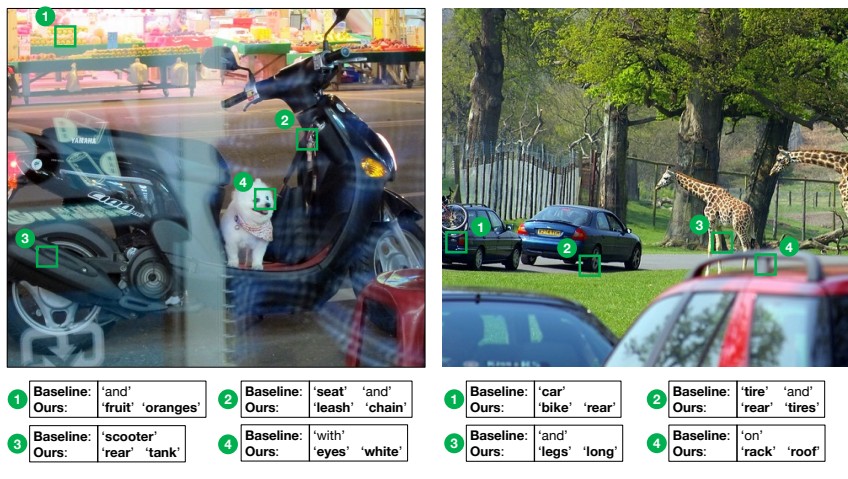

Figure 3: Output logit visualizations for (a) `Qwen2.5-3B` and (b) `-7B` models, comparing LLaViT with the baseline. For each visual token, we extract the top-3 words and display the two most sensible words, filtering irrelevant symbols such as '\n' or punctuations. Words are shown in bold if they are relevant to the corresponding image patch.

QKV parameters. These results validate that learning modality-specific QKV projections does indeed improve the visual representations within the LLM and their alignment to text representations. Then, adding bidirectional attention or local and global visual features further improves the performance on all benchmark categories. For example, incorporating bidirectional attention with separate QKV projections (row 3) improves separate QKV only (row 2) by 1.4pp on the OCR & Chart category for both the 3B and 7B models. Also, leveraging local and global visual (row 4) also has a profound effect, improving the performance of OCR & Chart by 1.4pp and 1.8pp for the 3B and 7B models. Finally, combining all three methods in LLaViT exhibits the strongest performance in all benchmark categories except Knowledge.

### 4.5 QUALITATIVE RESULTS

We find that improving the LLM's ability to process visual information also leads to better translation of visual tokens, as discussed in Section 2.2. Figure 3 visualizes the translations of two images, one comparing LLaViT with the baseline on `Qwen2.5-3B` (left) and another on `Qwen2.5-7B` (right). Note that we applied some post-processing to the visualizations in Figure 3 for the sake of better readability, but present more detailed visualizations in the Appendix.

On both images, we first note that the baseline model is able to correctly capture high-level concepts, such as "scooter" and "dog" for Figure 3(a), and "car", "tire", "gir(affe)", "roof" in Figure 3(b). However, LLaViT is able to go beyond high-level concepts to capture more diverse and fine-grained details, such as "fruits", "leash", "rear", "eyes" in Figure 3(a), and "bike", "rear", "legs", "long", and "rack" in Figure 3(b). These examples clearly show that, when applying our method, the LLM is able to better understand and translate visual concepts, which is likely correlated to the significant performance gains we observe in evaluation benchmarks.

## 5 CONCLUSION

We presented LLaViT, a novel architecture that enhances multimodal large language models by improving the LLM's ability to process visual information. We proposed three simple yet effective techniques: (i) learning separate QKV projection parameters for attention layers within the LLM, (ii) enabling bidirectional attention on visual tokens, and (iii) using local and global visual features. Our experiments and evaluations on a wide range of LLMs and benchmarks clearly demonstrate the versatility and effectiveness of our approach. We believe our work has provided a new perspective on MLLM design, and hope that others can build upon our findings to further improve MLLMs.

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

# A    MEMORY AND COMPUTE EFFICIENCY OF LLaViT

Table 4: Number of parameters (in Billion) of the entire model (*i.e.*, vision encoder $g$, projector $f_\phi$, and LLM $h_\theta$), comparing LLaViT with the baseline on various base LLMs.

| Base LLM | Method | Num. Params (B) | $\Delta\%$ |
|---|---|---|---|
| Qwen2.5-1.5B | Baseline
Ours | 1.85
1.94 | + 4.9 |
| Qwen2.5-3B | Baseline
Ours | 3.40
3.59 | + 5.7 |
| Qwen2.5-7B | Baseline
Ours | 7.93
8.40 | + 5.9 |
| Qwen2.5-14B | Baseline
Ours | 15.10
16.87 | + 11.7 |

## A.1    MEMORY

A core component of LLaViT is learning separate QKV projections in all layers of the LLM. We provide details of the parameter counts when comparing the baseline and LLaViT in Table 4. Although learning separate QKV projections increases the model's total number of parameters, the additional parameters only account for a 5%~12% of the entire parameter count. Despite the relatively small increase in number of parameters, we observed a much more significant increase in performance, where the 3B LLaViT outperformed 7B baseline ($\geq 2\times$ parameters) and was even competitive with the 14B baseline ($\geq 4\times$ parameters) on Vision Centric and OCR&Chart benchmarks. From a different perspective, we could argue that applying LLaViT allows practitioners to maintain (or even improve) the performance of the MLLM while reducing the parameter count by more than half. Thus, the small increase in parameters is a great tradeoff for significantly improved performance.

Table 5: Analysis of floating-point operations (FLOPs), comparing the baseline and LLaViT using Qwen2.5-3B as the base LLM.

| Module | FLOPs (GFLOPs) |
|---|---|
| Causal Attention (*per attention layer*) | 38.7 |
| Causal + Bidirectional Attention (*per attention layer*) | 40.8 |
| MLP Block (*identical for both baseline and LLaViT*) | 92.4 |
| LM Head (*identical for both baseline and LLaViT*) | 632.7 |
| Entire LLM for Baseline (*36 layers of Attention & MLP + LM Head*) | 5348.7 |
| Entire LLM for LLaViT (*36 layers of Attention & MLP + LM Head*) | 5426.1 |

## A.2    COMPUTE

Learning separate QKV projections does **not** increase the total FLOPS of the model, because the LLM simply routes a subset of the tokens (*i.e.* visual tokens) to $\{W_Q^{\text{vis}}, W_K^{\text{vis}}, W_V^{\text{vis}}\}$ instead of $\{W_Q^{\text{text}}, W_K^{\text{text}}, W_V^{\text{text}}\}$. This is similar to the Mixture-of-Experts Shazeer et al. (2017) mechanism, where inputs are gated to a specific expert. In our case, we use the token's modality to determine which QKV projection to use instead of a separate routing network.

For local-global features, there is a trivial increase in computation. This is because we concatenate the visual features in the feature dimension, so the only difference is that a single MLP projection has a larger input dimension. Furthermore, we extract visual features from different layers of the same CLIP vision encoder, which means that the local and global features can be extracted from a single forward pass of the vision encoder.

Table 6: Evaluation results on 17 MLLM benchmarks using `Phi-3.5-mini` Abdin et al. (2024) as the base LLM. "Baseline" refers to LLaVA-1.5 trained with the respective LLMs. We present evaluation results on individual benchmarks, as well as the averages of each category, colored in blue.

| Method | Vision Centric | | | OCR & Chart | | | | | | Knowledge | | | General | | | | | | | | |
| | RealWorldQA | MMVP | Avg | ChartQA | DocVQA | InfoVQA | OCRBench | TextVQA | Avg | SciQA | AI2D | Avg | GQA | MMBench-EN | MMBench-CN | MME-P | POPE | VizWiz | MMStar | VQAv2 | Avg |
|---|---|---|---|---|---|---|---|---|---|---|---|---|---|---|---|---|---|---|---|---|---|
| | | | | | | | | | Phi-3.5-mini | | | | | | | | | | | | |
| Baseline | 56.2 | 30.7 | 43.5 | 21.5 | 25.2 | 23.9 | 33.9 | 45.1 | 29.9 | 73.5 | 65.0 | 69.3 | 61.5 | **71.7** | 63.5 | 1449.4 | **86.0** | 40.2 | 40.0 | 76.3 | 64.0 |
| LLaViT | **57.3** | **41.3** | **49.3** | **23.2** | **28.8** | **25.7** | **38.0** | **48.5** | **32.9** | **74.4** | **67.8** | **71.1** | **63.2** | 70.4 | **65.0** | **1483.3** | 85.9 | **43.7** | **42.8** | **77.8** | **65.4** |

| Method | Vision Centric | | | OCR & Chart | | | | | | Knowledge | | | General | | | | | | | | |
| | RealWorldQA | MMVP | Avg | ChartQA | DocVQA | InfoVQA | OCRBench | TextVQA | Avg | SciQA | AI2D | Avg | GQA | MMBench-EN | MMBench-CN | MME-P | POPE | VizWiz | MMStar | VQAv2 | Avg |
|---|---|---|---|---|---|---|---|---|---|---|---|---|---|---|---|---|---|---|---|---|---|
| | | | | | | | | | Qwen2.5-3B | | | | | | | | | | | | |
| Baseline | 54.5 | 24.0 | 39.3 | 17.6 | 22.8 | 22.0 | 32.3 | 42.4 | 27.4 | 73.4 | 61.6 | 67.5 | 61.3 | 71.0 | 66.6 | 1451.7 | 85.8 | 49.7 | 44.0 | 75.9 | 65.9 |
| No Vis. Attn. | 41.7 | 8.0 | 24.9 | 10.6 | 9.1 | 18.8 | 3.7 | 11.6 | 10.7 | 71.0 | 58.2 | 64.6 | 49.0 | 46.0 | 43.6 | 1059.7 | 78.6 | 38.3 | 34.9 | 58.0 | 50.2 |
| | | | | | | | | | Qwen2.5-7B | | | | | | | | | | | | |
| Baseline | 58.7 | 31.3 | 45.0 | 23.0 | 27.0 | 24.4 | 36.3 | 48.1 | 31.8 | 75.7 | 67.6 | 71.7 | 63.2 | 71.1 | 68.0 | 1506.6 | 87.1 | 58.7 | 46.2 | 78.4 | 68.5 |
| No Vis. Attn. | 56.3 | 25.3 | 40.8 | 17.2 | 21.3 | 22.0 | 32.1 | 45.2 | 27.6 | 75.6 | 64.0 | 69.8 | 62.8 | 71.0 | 67.9 | 1489.4 | 86.6 | 54.3 | 41.8 | 77.3 | 67.0 |

Table 7: Comparison of LLaVA-1.5 with and without visual attention. This table shows all the individual benchmark scores and corresponds to Table 1 in the main paper.

Finally, we analyze the computation cost of bidirectional attention for visual tokens. The key difference to consider here is a purely causal mask versus the causal + bidirectional attention mask, as shown in Figure 2(b). Since the exact FLOPS depends on the number of input tokens, we consider the case where the total sequence length is 1024, 576 of which are visual tokens (448 text tokens). We present an analysis of FLOPTS using Qwen2.5-3B as the base LLM in Table 5.

## B  ADDITIONAL RESULTS AND FULL TABLES

**Results on Phi-3.5**   To demonstrate the effectiveness of LLaViT on diverse LLM architectures, we also experiment with `Phi-3.5-mini` Abdin et al. (2024) as the base LLM and present the results in Table 6. The trends on `Phi-3.5-mini` are consistent with the trends seen on the `Qwen2.5` family of LLMs: we observe significant gains in all benchmark categories, especially in Vision Centric and OCR & Chart, where LLaViT improves the baseline by 5.8pp and 3.0pp, respectively. Also, on the Knowledge and General categories, we see improvements of 1.8pp and 1.4pp, respectively.

**Full Table for Table 1**   In Table 1 of our main paper, we presented a condensed table to demonstrate the effect of removing visual attention from the LLM. We present the full version with all individual benchmarks in Table 7.

# C ADDITIONAL QUALITATIVE RESULTS

We present some additional qualitative results below.

## C.1 DETAILED VERSION OF FIGURE 1

Figure 4 is a detailed visualization corresponding to Figure 1 in the main paper. This Figure visualizes how the LLM interprets visual tokens at the input and output layers of the LLM. We visualize select tokens and display the Top-3 words alongside their cosine similarities or output probabilities.

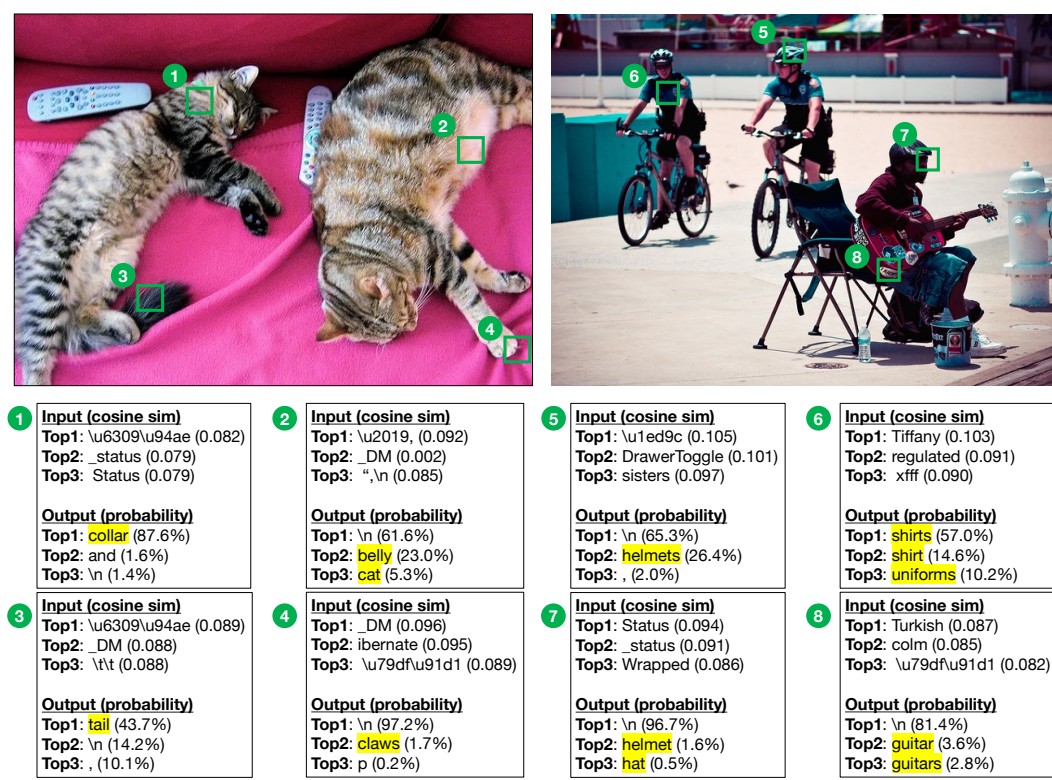

Figure 4: Detailed version of Figure 1, showing the raw Top-3 input and output words, alongside their cosine similarity or output probabilities. Non-alphabetic characters are displayed in their unicode representations, and we highlight semantically relevant words in yellow.

## C.2 ADDITIONAL OUTPUT LOGIT VISUALIZATIONS

Figures 5 and 6 illustrate the translation of visual tokens at the output of the LLM when using `Qwen-2.5-3B` and `-7B` as the base LLM, respectively. For both figures, we display the top-3 words and their probabilities for both LLaViT and the baseline, and highlight semantically relevant words in yellow.

We notice some striking differences when comparing LLaViT with the baseline on both the `3B` and `7B` LLMs. First, the baseline model often predicts the newline token ('\n') with high probability, which is an artifact of the training data where the newline token always follows the last image token. This effect seems to be largely mitigated in LLaViT. Second, while the baseline model is correctly translate visual tokens to semantically related words (*e.g.*, "seat", "scooter", "railing", "post" in Figure 5 and "pens", "logo", "tag", "car", "tire" in Figure 6), LLaViT exhibits a higher level of fine-grained detail. Some examples of are:

- **The specific type of object:** "oranges", "tank", "lamp", "metal" (Figure 5) and "label", "rack", "roof" (Figure 6).

- **Relative position of objects:** "side" and "rear" in both Figures.
- **Text within the image:** "pay" (Figure 5) and "Guiness" (Figure 6).
- **Objects not identified by the baseline model:** "fruits", "leash" (Figure 5) and "string", "rack" (Figure 6).

This demonstrates a stronger alignment of vision and text when the LLM can simultaneously serve as an extended vision encoder, as in LLaViT.

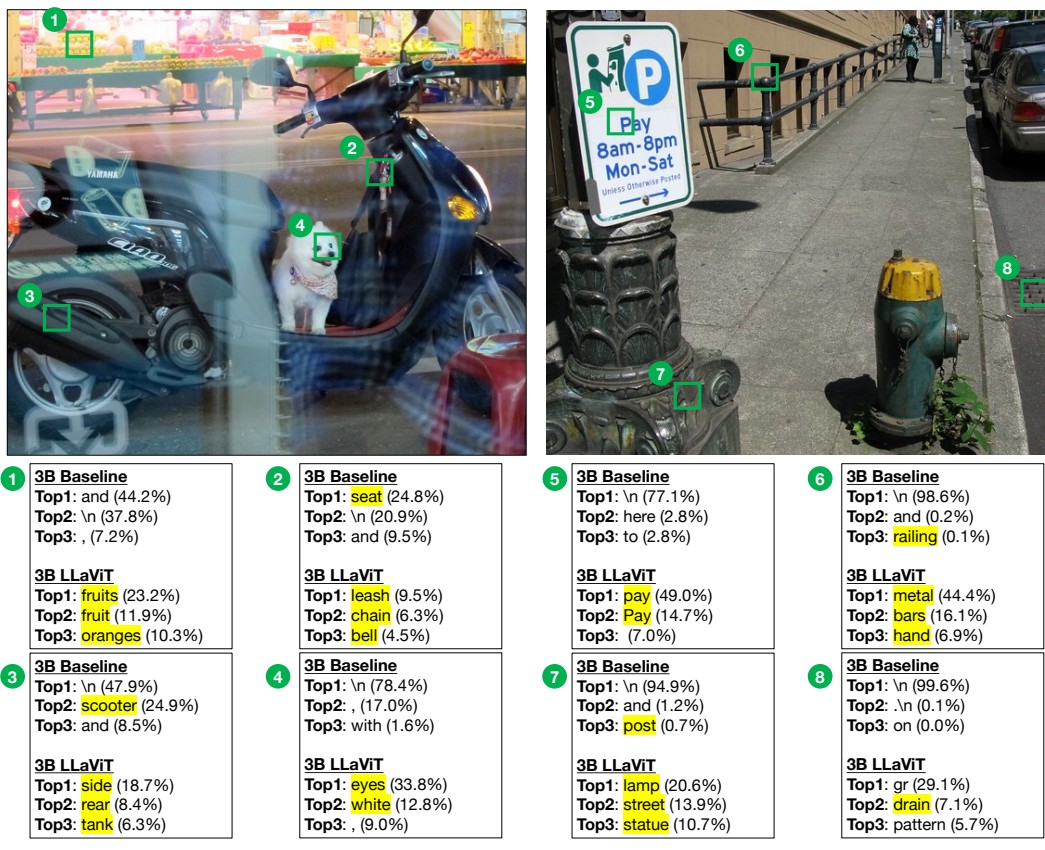

Figure 5: Visualizing the output of visual tokens, comparing the baseline and LLaViT using `Qwen2.5-3B` as the base LLM. We visualize the top-3 outputs for each model alongside the output probabilities, and highlight semantically relevant words in yellow.

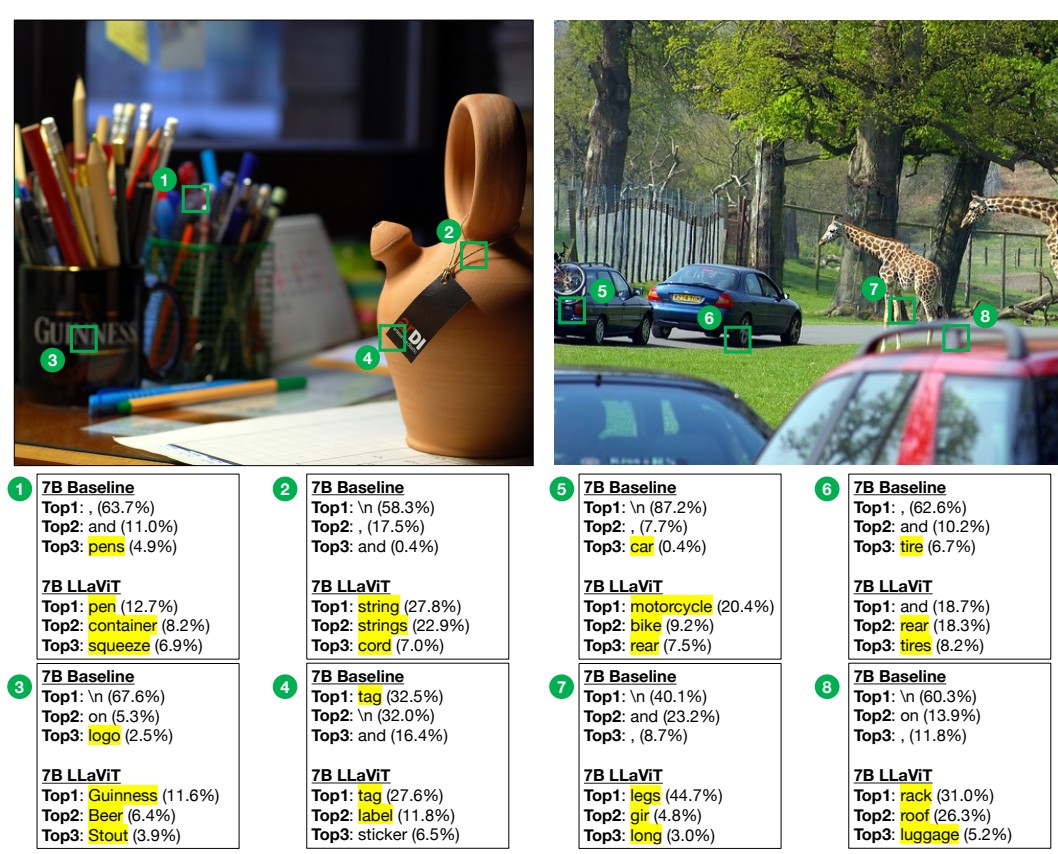

Figure 6: Visualizing the output of visual tokens, comparing the baseline and LLaViT using Qwen2.5-7B as the base LLM. We visualize the top-3 outputs for each model alongside the output probabilities, and highlight semantically relevant words in yellow.

## D    RELATED WORKS

**Vision Language Models.**    Vision language models (VLMs) have evolved rapidly, enabling a wide range of tasks such as zero-shot image classification Radford et al. (2021); Zhai et al. (2023), image captioning Li et al. (2022); Xu et al. (2016); Karpathy & Fei-Fei (2015), open-world detection Liu et al. (2023b); Minderer et al. (2022), and even multimodal search Vo et al. (2019); Chen & Bazzani (2020); Lee et al. (2021); Zhu et al. (2024). More recently, multimodal large language models (MLLMs) architectures Dai et al. (2023); Liu et al. (2023a; 2024) have emerged as a prominent type of VLMs, leveraging the power and versatility of LLMs.

**Improving LLaVA.**    Following the success of LLaVA Liu et al. (2023a; 2024), many works have aimed to improve the LLaVA architecture from various perspectives. Firstly, many previous works train more capable and versatile vision encoders Chen et al. (2024a); Ranzinger et al. (2024); Wang et al. (2024) that improve performance by enhancing the visual features fed into the LLM. For example, AM-RADIO Ranzinger et al. (2024) leverage multi-teacher distillation to create a single vision foundation model that inherits the capabilities of all teacher models, and Qwen2.5-VL proposes a new vision transformer that can process images efficiently at native and dynamic resolutions, which allows the MLLM to better process images of high-resolution and irregular aspect ratios. Other works focus on improving the vision-language projector design. For example, Honeybee Cha et al. (2024) proposes the D-Abstractor which leverages deformable attention Zhu et al. (2020) to reduce the number of visual tokens while preserving locality, and Cambrian-1 Tong et al. (2024a) proposes a spatial vision aggregator that aggregates features from multiple vision encoders. Another line of work focuses on improving the quality of multimodal instruction data. PixMo Deitke et al. (2024) introduces a collection of datasets that include high-quality human annotated captions, pointing and counting datasets, as well as some synthetic datasets that can be used to train more capable MLLMs. In addition, ShareGPT4V Chen et al. (2024b) and LVIS-INSTRUCT4V Wang et al. (2023a) collect high-quality vision-language datasets by carefully prompting and curating responses from GPT-4V OpenAI (2023).

Compared to the previous works to improve LLaVA-like MLLMs, we introduce a new paradigm for MLLM architecture design where the LLM itself serves as an extension of the vision encoder, rather than just a language model. The three key modifications we introduced are a carefully selected combination designed to validate this new paradigm. While these components may have appeared in isolation in other contexts with different goals Yao et al. (2024); Wang et al. (2023b), their combination and integration within our framework serves a novel purpose: to progressively enrich visual representations inside the LLM. The strong and consistent improvements across 17 benchmarks, as well as the qualitative improvements shown in Figures 3, 5, and 6, demonstrate the effectiveness of this new architectural design. We believe that this paradigm of "the LLM as a visual encoder" opens up a promising avenue for future research. By extending the visual processing into the LLM, we can leverage the powerful architectural innovations from both vision and language model research, and hope that our work will inspire the community to explore this direction further.

## E    LIMITATIONS

A limitation of LLaViT is that it incurs a slight increase in model parameters by learning separate QKV projections for visual tokens. However, as discussed in Section A, the benefits outweigh the cost, since adding these parameters provides enough performance gain to even outperform models with double the size. From a different perspective, this approach means a smaller base LLM can be utilized to achieve the same level of performance, making the modest parameter increase a worthwhile investment.

