# OpenReview forum: "Rethinking Visual Information Processing in Multimodal LLMs"
_ICLR.cc/2026/Conference — Submitted to ICLR 2026_

### Official Review · Reviewer_8GkR · 2025-10-27

**Soundness:** 3
**Presentation:** 2
**Contribution:** 2
**Rating:** 4
**Confidence:** 4

**Summary:**

This work proposes to improve the visual understanding capabilities of encoder-decoder large vision-language models through applying architectural enhancements over the LLM part. In particular, considering LLaVA as the baseline, the work proposes to learn separate QKV projection layers for the visual tokens as they are propagating through the LLM, and supplement this approach by utilizing a hybrid attention mechanism while also utilizing channel-wise concatenated visual tokens from different blocks of the visual encoder. Through experimental evaluation, the work aims to highlight the usefulness of the proposed methodology.

**Strengths:**

Below are the primary strengths of the work:

- The exact methodology, i.e. the combination of adding new QKV projections layers for visual tokens, considering visual tokens from different visual encoder blocks and applying bidirectional attention in this way on encoder-decoder VLMs is novel and interesting.
- The experiments demonstrate the desirable performance of the proposed method over the vanilla baselines, under a number of benchmarks with a number of models.
- The writing is clear and easy-to-follow.

**Weaknesses:**

Below are the primary weaknesses of the work:

**W1: Missing Citations in the Motivation Section:** The paper's primary motivation stems from the different approaches to token-level processing for visual versus textual data. While the analysis presented to support this motivation is sound, it fails to cite several critical works that have conducted similar explorations. Specifically:

- The paper uses the term "modality gap" without citing the original works that introduced it [A, B]. Although those studies may focus on contrastive learning, their findings are foundational to the exploration in this paper and require attribution.

- More importantly, the paper overlooks the work of [C], which had already explored and presented the majority of the findings discussed in the motivation section.

While I understand this analysis is intended to serve as motivation, in its current form, the text fails to establish these crucial links to the existing literature. Accordingly, if the explorations of the authors provide an orthogonal direction to the existing literature, e.g. over [C], the authors should clarify this in the text explicitly.

**W2: Ambiguities in Technical Contributions:** The authors have acknowledged (Lines 1002-1004 of Appendix D) that their methodology is not entirely novel, but rather is a _unique_ combination of existing methods, namely the additional QKV layers for visual tokens, bidirectional attention for visual tokens and the utilization of features from different visual encoder blocks. This is appreciated, though there are still several missing links to the existing literature, making the exact contributions of the work ambiguous. In particular:

- The idea of utilizing additional vision-specific parameters _within the LLM_ is explored quite well in the literature of monolithic VLMs [D, E, F, G, H], though the work claims to propose a "new paradigm" on this end. Notably,  EVEv1.5 [D] and Mono-InternVL [E] realize this idea through separating the FFN blocks, EVEv2 [F] does so by separating both the FFN and LayerNorm blocks and VoRA [G] realizes this goal by utilizing LoRA projections specifically for vision.  Although I acknowledge the novelty of the QKV aspect, I still believe that there should be at least a verbal discussion comparing LLaViT to these works with experimental comparison against them being the ideal and fair case.
- Using hybrid attention masking to allow visual tokens to attend each other bidirectionally while keeping the causal mask for textual tokens is a well-known practice in the VLM literature [I, J]. Unfortunately the main text of the work misses this and the narrative seems to present this as a novel aspect of LLaViT.

**W3: Additional Parameters and Costs:** As LLaViT learns separate QKV projection matrices, this incurs an unavoidable cost over the baseline LLaVA. Notably, this addition in parameters is included _on top of the existing visual encoder_. This might limit the future usage of the work for several memory or inference time constrained settings.

---
[A] Liang, V. W., Zhang, Y., Kwon, Y., Yeung, S., & Zou, J. Y. (2022). Mind the gap: Understanding the modality gap in multi-modal contrastive representation learning. Advances in Neural Information Processing Systems, 35, 17612-17625.

[B] Schrodi, S., Hoffmann, D. T., Argus, M., Fischer, V., & Brox, T. (2024). Two effects, one trigger: On the modality gap, object bias, and information imbalance in contrastive vision-language representation learning. arXiv preprint arXiv:2404.07983.

[C] Shukor, M., & Cord, M. (2024). Implicit multimodal alignment: On the generalization of frozen llms to multimodal inputs. Advances in Neural Information Processing Systems, 37, 130848-130886.

[D] Diao, H., Cui, Y., Li, X., Wang, Y., Lu, H., & Wang, X. (2024). Unveiling encoder-free vision-language models. Advances in Neural Information Processing Systems, 37, 52545-52567.

[E] Luo, G., Yang, X., Dou, W., Wang, Z., Liu, J., Dai, J., ... & Zhu, X. (2025). Mono-internvl: Pushing the boundaries of monolithic multimodal large language models with endogenous visual pre-training. In Proceedings of the Computer Vision and Pattern Recognition Conference (pp. 24960-24971).

[F] Diao, H., Li, X., Cui, Y., Wang, Y., Deng, H., Pan, T., ... & Wang, X. (2025). Evev2: Improved baselines for encoder-free vision-language models. arXiv preprint arXiv:2502.06788.

[G] Wang, H., Ye, Y., Li, B., Nie, Y., Lu, J., Tang, J., ... & Huang, C. (2025). Vision as lora. arXiv preprint arXiv:2503.20680.

[H] Bavishi, R., Elsen, E., Hawthorne, C., Nye, M., Odena, A., Somani, A., & Tasırlar, S. (2023). Introducing our multimodal models. Adept Blog.

[I] Beyer, L., Steiner, A., Pinto, A. S., Kolesnikov, A., Wang, X., Salz, D., ... & Zhai, X. (2024). Paligemma: A versatile 3b vlm for transfer. arXiv preprint arXiv:2407.07726.

[J] Steiner, A., Pinto, A. S., Tschannen, M., Keysers, D., Wang, X., Bitton, Y., ... & Zhai, X. (2024). Paligemma 2: A family of versatile vlms for transfer. arXiv preprint arXiv:2412.03555.

**Questions:**

In addition to the potential responses to the weaknesses I raised above, I have the following questions:

- Can you please clarify why only QKV projections were learned specifically for visual tokens? Why not separate the FFN blocks or LayerNorms blocks but only the QKV projection layers? How would LLaViT perform under these settings?
- Why consider the features from the 5th, 15th and 23rd blocks of the visual encoder? Why these block indices in particular? How does LLaViT work for different block choices?

---

> ### Author Response · Authors · 2025-11-20
> **Response to Reviewer 8GkR**
>
> Thank you for your time and effort in providing feedback on our paper. We truly appreciate the constructive criticisms and believe it to be extremely valuable in improving our paper. Below, we answer questions and address the weaknesses raised.
>
> ## Missing citations in motivation
>
> We thank you for pointing out important relevant works that we failed to properly acknowledge in our motivations. We will closely examine the cited works and update the manuscript accordingly in the upcoming days.
>
> ## On technical contributions
>
> We believe that the core novelty of our paper lies not in the invention of these individual components, but in the new paradigm we propose for MLLMs. This paradigm is motivated by the LLM’s tendency to translate visual to tokens into text (without any explicit supervision), and reframes the role of the LLM from passively receiving pre-processed visual features to actively aiding the visual encoding process.
>
> The three key modifications we introduced - separate QKV projections for visual tokens, bidirectional attention, and local/global visual features - are a carefully selected combination designed to **validate** this new paradigm. While these components may have appeared in isolation in other contexts with different goals, their integration within our framework serves a novel purpose: to progressively enrich visual representations inside the LLM. The strong and consistent improvements we demonstrate across 17 benchmarks demonstrate the effectiveness of this new architectural design.
>
> We believe that this paradigm of "the LLM as a visual encoder" opens up a promising avenue for future research. By extending the visual processing into the LLM, we can leverage the powerful architectural innovations from both vision and language model research. We hope our work will inspire the community to explore this direction further. Some potential future work following this paradigm could include:
>
> - Leveraging efficient attention mechanisms: To improve the efficiency of processing high-resolution visual inputs, attention mechanisms from vision transformer research could be adapted for the visual tokens within the LLM. For example, the Shifting Window (Swin) attention mechanism could be applied exclusively to visual tokens.
> - Auxiliary losses for visual alignment: Our analysis showed that the LLM learns to translate visual tokens into meaningful text representations. This opens up the possibility of applying auxiliary losses directly to the output representations of visual tokens. For example, we could encourage these tokens to predict relevant textual descriptions, object tags, or spatial coordinates, thereby creating a stronger alignment signal to improve the overall quality of the learned visual features.
>
> ## Additional Parameters/Cost & Clarification on QKV projections
>
> We provide a comprehensive analysis of the memory and computational overhead of LLaViT in Tables 4 and 5 (Appendix). To summarize our analysis, LLaViT only incurs a 5~12% increase in parameters (depending on the LLM) and only increases FLOPS by roughly 1.5% compared to the baseline. Please refer to Section A in the Appendix for the full analysis.
>
> The reason we only learn separate QKV projections for visual tokens (rather than FFN blocks) is closely tied to efficiency. As mentioned previously, the parameter increase of LLaViT is relatively minor specifically because we only opt to learn separate QKV projections for visual tokens. If the FFN layers are also separated, it would be similar to running two LLMs in parallel; one for the visual tokens and another for the text tokens, with attention spanning across the two LLMs. In this case, we cannot guarantee that the efficiency-accuracy tradeoff would be worthwhile. In our case, however, we assert that the performance improvement is quite significant while the loss in efficiency is minimal.
>
> ## Choice of visual features
>
> We admit that the features from the 5th, 15th, and 23rd blocks of the vision encoder were based on heuristics rather than concrete evidence. The motivation was to employ a diverse set of features: one from an early layer, another from a middle, and another close to the final layer. This aligns with the common notion that visual features from early layers are more focused on local representations and become more global towards the end of the vision encoder.
>
> With that said, a recent work [A] performed a comprehensive analysis of using outputs from different layers of the vision encoder and found that the best output for downstream tasks is **not necessarily** the final output, but rather an intermediate output. However, which specific layer produces the strongest downstream result varies from task to task. In our case, we ensemble the output from multiple different layers such that the LLM can effectively modulate the different visual representations.
>
> [A] Bolya, D., et. al., (2025). Perception Encoder: The best visual embeddings are not at the output of the network

---

> > ### Comment · Reviewer_8GkR · 2025-11-25
> >
> > I thank the authors for their response. Although the manuscript is yet to be updated, I acknowledge the authors' promise to update it accordingly with the missing citations.
> >
> > Regarding the more significant weakness that I raised on comparing LLaViT with existing monolithic VLMs (which actually aim to do exactly what the authors have stated the _new paradigm that they are exploring_), the authors have yet to provide a comparison between their works and these existing works.
> >
> > I would like to note that although VoRA is not a published work and EVEv2 (ICCV 2025) was very recently published, EVE (NeurIPS 2024) and Mono-InternVL (CVPR 2025) are all peer-reviewed and published works reasonably long ago that have already explored the novel paradigm this work states it validates as a new paradigm. To reiterate the point from my initial review, I believe that it is imperative to have at least a strong and detailed verbal comparison against these works with experimental comparison against them being the ideal and fair case.
> >
> > Finally, the authors have stated that the particular blocks they have chosen are based on the analyses presented in the literature broadly. While I acknowledge that this is a valid initial point to explore, I believe that the claims of the work would be stronger if the work had at least some quantitative analysis behind this choice in the form of an ablation.
> >
> > Accordingly, given the unaddressed major concerns and weaknesses, my stance on the work is still closer to a rejection.

---

### Official Review · Reviewer_TRSK · 2025-10-29

**Soundness:** 3
**Presentation:** 3
**Contribution:** 3
**Rating:** 6
**Confidence:** 3

**Summary:**

The paper presents LLaViT, a new approach that rethinks visual information processing in multimodal LLMs (MLLMs). Instead of viewing the LLM as a purely linguistic component that receives aligned visual embeddings from a frozen vision encoder, the authors argue that the LLM itself performs substantial in-model translation of visual to textual representations. Based on this insight, they propose three architectural modifications that allow the LLM to act as an extended vision encoder:

1. Separate QKV projections for visual tokens — enabling modality-specific attention parameters to reduce misalignment between text and visual embeddings.
2. Bidirectional attention on visual tokens — removing causal masking within visual tokens to improve information flow.
3. Local-global visual features — concatenating multi-layer features from the vision encoder to provide both fine-grained and semantic representations.

These modifications are integrated into the LLaVA framework across multiple base LLMs (Qwen2.5 and Phi-3.5). On 17 multimodal benchmarks (e.g., GQA, TextVQA, RealWorldQA, MMVP), LLaViT consistently outperforms LLaVA-1.5 by a large margin. Ablation and visualization analyses show that the proposed mechanisms enhance visual token alignment, fine-grained comprehension, and visual-language consistency without major computational overhead.

**Strengths:**

- Clear conceptual motivation: reinterprets the LLM as part of the vision pipeline rather than a downstream consumer of features.
- Introduces three simple yet effective modifications, each empirically validated with clear ablations.
- Achieves substantial gains across diverse multimodal benchmarks compared to llava 1.5.
- Solid ablations on multiple LLM scales, both standard and high-resolution settings, and consistent methodology.
- Qualitative results provide intuitive evidence that visual tokens become more semantically aligned.

**Weaknesses:**

- The absolute performance remains relatively low compared to state-of-the-art models. While this is understandable given the limited dataset and compute budget, it raises questions about scalability. Can LLaViT be extended as a fine-tuning method for existing high-end MLLMs such as the Qwen-VL or InternVL series? Demonstrating compatibility with frontier models would substantially enhance the practical usability and relevance of this approach.

- The paper provides little discussion on the textual capabilities of LLaViT. It is well-known that multimodal fine-tuning often degrades the base LLM’s performance on pure-text tasks. Does the proposed method alleviate or exacerbate this issue? This is a minor concern and I understand if the author can't have results during the rebuttal.

- The experimental comparisons are limited to the LLaVA family. Can the proposed architecture generalize to other MLLM paradigms such as CogVLM, Flamingo, or BLIP-2 style frameworks? Since the modifications are largely architectural, validating cross-framework applicability would strengthen the paper’s generality.

- The ablation analysis lacks coverage of different vision encoders. The current experiments focus primarily on CLIP-H as the encoder. Additional results with other backbones (e.g., SigLIP2, or DINOv2) or even different scales of the same encoder would help confirm that the proposed mechanisms are not specific to a single visual backbone.

**Questions:**

See Weakness.

I would recommend a borderline acceptance at this stage. The paper presents clear insights and solid empirical evidence, but its current scope is confined to LLaVA-style architectures. I would be inclined to raise the score if the authors can demonstrate the generalizability of LLaViT to pretrained MLLMs and to other architectures beyond the LLaVA family, such as CogVLM or Flamingo.

---

> ### Author Response · Authors · 2025-11-20
> **Response to Reviewer TRSK**
>
> Thank you for your time and effort in providing valuable feedback on our paper. We are pleased to hear the strengths of our paper, and appreciate the constructive criticisms. Below, we answer questions and address the weaknesses raised.
>
> ## Comparison with SOTA
>
> Since LLaViT focuses on architectural improvements to MLLMs in general, we wanted to ensure that the experimental setting was controlled so that the effects of LLaViT would be isolated from external factors, such as quality/quantity of data, base LLM, etc. Thus, we felt that it was difficult to directly compare with recent SOTA MLLMS which use significantly more training data and computational resources. Unfortunately, scaling up data and computational resources is not something we can achieve with our resources. While the lack of such direct comparisons could be a limitation of the paper, we believe that our strong results compared to the baseline (conducted under identical experimental settings) validate the effectiveness of LLaViT.
>
> We believe LLaViT (and the idea of using the LLM as an extended vision encoder) can be applied to any general MLLM that uses visual tokens as inputs to a pre-trained LLM. However, because LLaViT introduces an architectural change, it may be difficult to apply our idea in the finetuning stage. Rather, it would be more appropriate to apply this idea when training a new MLLM, starting with a pre-trained LLM. In addition, we’d like to note that the three proposed modifications are means to validate our newly proposed paradigm of using LLM as part of the vision pipeline. In fact, this paradigm of "the LLM as a visual encoder" could open up a promising avenue for future research, which could be adopted by future iterations of large-scale MLLMs such as Qwen-VL or InternVL.
>
> ## Textual capabilities of LLaViT
>
> We posit that multimodal fine-tuning degrades the LLM’s performance on pure-text tasks due to catastrophic forgetting, a phenomenon that is common in neural networks when trained on a new dataset. Specifically, multimodal fine-tuning is often done with small amounts of pure text data, which is the major contributor to forgetting. While our separate QKV projections could alleviate this effect in the attention layers, other layers of the LLM (e.g., MLP layers) would still undergo catastrophic forgetting. Thus, we do not expect LLaViT to significantly alleviate this issue, although it would definitely not exacerbate it either.
>
> ## Extending to other MLLMs
>
> Our method and newly proposed paradigm pertain to families of VLMs/MLLMs that input the visual tokens directly into the LLM. While we used LLaVA in our experiments, our method and paradigm could apply to other recent models such as Qwen-VL or InternVL. However, this does not necessarily apply to models such as Flamingo or BLIP-2, which do not feed the visual tokens directly into the LLM much like text tokens.
>
> ## Ablation on vision encoders
>
> We are confident that our method will extend well to other vision encoders, as the methodology we proposed mostly focuses on how the LLM processes visual tokens. Unfortunately, we are currently unable to access enough computational resources to conduct experiments with a different vision backbone. We hope the reviewer will understand.

---

> > ### Author Response · Authors · 2025-11-20
> > **Response to Reviewer Afru [2/2]**
> >
> > ## Adaptive QKV routing
> >
> > Yes, and in fact, the separate QKV projections could be thought of as a special case of MoE on the QKV projection parameters of the LLM, where routing is dependent on the input modality. Using a learned routing could work as well, but separating QKV projection parameters based on the input modality aligns closely with the motivation of our work, which is to reframe the LLM as an extension of the vision encoder.

---

> > ### Comment · Reviewer_TRSK · 2025-11-26
> > **Response to Authors**
> >
> > I thank the authors for the timely and detailed response.
> >
> > I understand that it is unrealistic to run additional experiments on other high-compute models such as Qwen-VL. However, I also agree with reviewer Afru that *relying solely on LLaVA-1.5 as the baseline is a substantial limitation for an ICLR 2026 paper*. I recognize that applying the proposed method to other styles of MLLMs is beyond the scope of the current work, and I agree with the authors that the method can likely be extended to other vision encoders. Nevertheless, without quantitative experiments, it is difficult to assess the generalizability of the approach.
> >
> > Therefore, I will keep my score unchanged.

---

### Official Review · Reviewer_Afru · 2025-10-31

**Soundness:** 2
**Presentation:** 3
**Contribution:** 3
**Rating:** 4
**Confidence:** 3

**Summary:**

This paper introduces LLaViT (Large Language Models as extended Vision Transformers), a novel framework that rethinks the role of LLMs in multimodal systems by enabling them to function as vision encoders through three key modifications:
(1) learning separate QKV projections for visual tokens,
(2) enabling bidirectional attention on visual tokens, and
(3) incorporating local and global visual features.
The authors argue that conventional LLaVA-like architectures suffer from a modality mismatch, where visual tokens are not well-aligned with the LLM's input space, and demonstrate that enhancing visual processing within the LLM leads to significant gains. Experiments across 17 benchmarks show that LLaViT outperforms LLaVA baselines, with a 3B model even surpassing 7B and 14B baselines on vision-centric tasks. Contributions include a new perspective on MLLM design, empirical validation of visual token translation in LLMs, and scalable improvements across diverse settings.

**Strengths:**

1. Originality: Reframing LLMs as vision encoders is genuinely novel. The three targeted modifications collectively close key visual-processing gaps, and the “visual token translation” mechanism (Sec. 2.2) offers fresh insight into MLLM internals.
2. Quality: Evaluation is comprehensive—17 benchmarks, multiple LLM families (Qwen2.5, Phi-3.5), and both Standard/Any-Res settings. Ablations (Tab. 3) and qualitative analyses (Figs. 3, 5–6) rigorously validate each component.
3. Clarity: The exposition is accessible and well-structured, with clear figures and tables; the motivation in Secs. 2.2–2.3 leads naturally to the methodological choices.
4. Significance: LLaViT achieves notable efficiency—smaller models outperform larger baselines—making it compelling for resource-constrained deployments, while the unified framework lowers integration costs and encourages broader adoption.

**Weaknesses:**

1. Baseline diversity: Comparisons stop at LLaVA-1.5; missing head-to-head results with recent MLLMs (e.g., Qwen-VL, InternVL) undercut claims of state-of-the-art performance.
2. Computational overhead: While parameter growth (5–12%, Tab. 4) and FLOPs (Tab. 5) are reported, end-to-end latency, peak memory, and throughput under realistic batch sizes/resolutions are not quantified—crucial for deployment.
3. Theoretical grounding: The benefit of semantic alignment is asserted but not explained; “visual token translation” remains descriptive without formal modeling (e.g., information-theoretic or perceptual analyses).
4. Data efficiency: Training hinges on high-quality PixMo-Cap; robustness in low-resource or noisy settings is untested.

**Questions:**

1. Generalization to video/3D: How would LLaViT extend to video or 3D? What architectural changes are needed to model temporal/spatial structure (e.g., frame tokenization, temporal pos. encodings, cross-frame attention, or 3D-aware token translation)?
2. Adaptive QKV routing: Could the separate Q/K/V projections be made input-adaptive (e.g., gating or MoE) rather than statically modality-routed to handle mixed-modality inputs more flexibly?
3. Bidirectional attention rationale: Why does bidirectional attention particularly help visual tokens? Is the gain tied to absent intrinsic ordering, and would similar effects appear in other set-structured modalities (e.g., point clouds)?
4. Parameter efficiency vs. performance: What are the accuracy–efficiency trade-offs? Could weight sharing, low-rank adapters, or pruning reduce overhead while preserving gains?

---

> ### Author Response · Authors · 2025-11-20
> **Response to Reviewer Afru [1/2]**
>
> Thank you for your time and effort in providing valuable feedback on our paper. We are pleased to hear that you found our paper to be clear, original, and of high quality. Below, we answer questions and address the weaknesses raised.
>
> ## Baseline Diversity
>
> Since LLaViT focuses on architectural improvements to MLLMs in general, we wanted to ensure that the experimental setting was controlled so that the effects of LLaViT would be isolated from external factors, such as quality/quantity of data, base LLM, etc. Thus, we felt that it was difficult to directly compare with recent SOTA MLLMS which use significantly more training data and computational resources. Unfortunately, scaling up data and computational resources is not something we can achieve with our resources. While the lack of such direct comparisons could be a limitation of the paper, we believe that our strong results compared to the baseline (conducted under identical experimental settings) validate the effectiveness of LLaViT.
>
> ## Computational Overhead and Efficiency & Tradeoff
>
> We agree that latency, peak memory, and throughput (both training and inference) are crucial for deployment scenarios. Compared with the baseline, our LLaViT requires slightly more memory (5~12%, same as the increase in parameter count), while latency and throughput are unaffected. This is because
>
> 1. the additional FLOPS by LLaViT is minor (~1%), and
> 2. In LLMs (and MLLMs), latency and throughput are mostly dictated by autoregressive generation, i.e., the number of tokens generated. In contrast, LLaViT only affects image tokens which are only provided as inputs, and thus, can be processed by a single forward pass.
>
> Thus, while LLaViT incurs extremely minor computational overhead and efficiency, the performance improvement is quite significant.
>
> ## Theoretical grounding
>
> The emergence of visual token translation is an intriguing phenomenon that we observed during our analysis. While we do not have theoretical analyses for this phenomenon, we reasoned that such behavior must emerge without supervision because it helps the model better understand image tokens.
>
> In order to get a better sense of how much LLaViT improves visual token translation in MLLMs, we attempted to quantify the qualitative results in Figure 1. We describe the procedure and results below.
>
> **Procedure**: From the top-3 word representations for each visual patch on the input and output level, we obtain the set of unique words (after filtering special tokens). Using the LLM-as-a-Judge approach, we prompt Claude Sonnet 4.0 with the image and the list of words, and ask it to list all words that are relevant to the image. The table below shows the total counts of relevant words for Figure 1 Right (3 men):
>
> | Representation | Relevant Word Count |
> | --- | --- |
> | Input Word Representations | 8 |
> | Output Word Representations (Baseline) | 63 |
> | Output Word Representations (LLaViT) | 141 |
>
> As expected, there are only 8 input word representations that are deemed relevant, while there are 63 in the output. Moreover, our LLaViT model outputs 141 unique and relevant words, validating the improved diversity and detail that LLaViT exhibits (as discussed in Section 4.5: Qualitative Results).
>
> ## Rationale for Bidirectional Attention
>
> Yes, the rationale behind bidirectional attention for visual tokens is that there is no intrinsic ordering of visual tokens. Text tokens have an inherent ordering, which is why causal attention is used in LLMs. In contrast, visual tokens are simply unrolled to 1D from left to right, top to bottom, and there is no specific reason for this (other than simplicity of implementation). When we apply a causal attention mask on visual tokens, the top left patch cannot attend to any other image patches, while the bottom right patch can attend to all other patches. Such imbalance can negatively affect visual representation learning within the LLM, which is why we opted to apply bidirectional attention to visual tokens.
>
> ## Video/3D
>
> We believe the core ideas of LLaViT could definitely be applied to video or 3D inputs.
>
> In the case of video inputs, we could apply the same QKV separation technique to video tokens (separating the QKV parameters between visual and text tokens), as video tokens are also different modalities from text. Furthermore, for Bidirectional Attention, we could apply bidirectional attention to the visual tokens of each frame, while retaining causal attention in visual tokens across different frames to maintain temporal causality.
>
> The same idea can be applied to 3D data. 3D data is inherently different from text modality, so it would be beneficial to separate QKV parameters for 3D tokens in the LLM. Also, as mentioned in “Rationale for Bidirectional Attention”, 3D tokens do not have any intrinsic ordering where causal attention makes sense, so using bidirectional attention would be beneficial as well.

---

> > ### Comment · Reviewer_Afru · 2025-11-25
> >
> > I thank the authors for their detailed response and for taking the time to conduct the additional qualitative analysis using the "LLM-as-a-Judge" metric.
> >
> > 1) I completely understand the computational limitations you mentioned in the rebuttal. Training or even fine-tuning large models like Qwen-VL or InternVL is undeniably expensive, and I don’t expect every submission to reproduce all state-of-the-art baselines from scratch. That said, from a scientific perspective, depending only on LLaVA-1.5 as the baseline is a substantial limitation for an ICLR 2026 paper. The field has moved quickly, and stronger baselines have become the norm.
> >
> > 2) The additional experiment using Claude to quantify “relevant words” is a clever and illustrative way to show the visual token translation effect—I found it genuinely interesting. However, it’s still more of a heuristic observation than a true theoretical explanation of why this alignment happens at a mathematical level. It convincingly shows that the phenomenon is real, but it doesn’t really push our understanding of the underlying modality alignment dynamics any further.
> >
> > 3）I accept the authors' point that inference latency is dominated by autoregressive decoding, so the impact on "Time to First Token" might be manageable. However, the memory overhead (5-12%) is not trivial for edge deployment scenarios, which is often where smaller models (like the 3B ones tested) are most relevant.
> >
> > 4）I appreciate the authors' perspective that the separate QKV projections can be viewed as a "special case of MoE" where routing is statically determined by modality.
> >
> > Therefore, I will maintain my original score.

---

### Official Review · Reviewer_FBFx · 2025-11-01

**Soundness:** 2
**Presentation:** 2
**Contribution:** 1
**Rating:** 2
**Confidence:** 5

**Summary:**

This paper proposes a visual information re-weighting pipeline intended to emphasize key regions in image representations and improve downstream recognition tasks. The framework is evaluated across several benchmarks to demonstrate performance improvements.

**Strengths:**

1. The paper is clearly written, and the structure is easy to follow.

2. The overall computational pipeline is straightforward to implement.

3. The motivation of improving visual emphasis and token weighting is generally relevant.

**Weaknesses:**

1. The claimed contribution based on the feature re-weighting strategy reads more like an intuitive design choice rather than a meaningful research idea.

2. The approach resembles common practices in project work, adjusting feature emphasis and reporting slightly improved accuracy without introducing new perspectives on the problem.

3. The paper does not discuss several closely related token aggregation approaches (e.g., DeepStack) and decouple transformers (e.g., MoT, EVEv2, Bagel, Mono-Internvl1.5). It is difficult to assess whether the proposed method has an obvious technical difference.

4. A major concern is that most of the compared baselines are far behind current state-of-the-art VLMs.

5. Only a small set of models and datasets are evaluated. Without comparison to modern, competitive baselines, it is unclear whether the method scales or holds advantage in realistic scenarios.

**Questions:**

1. How does this method differ in principle from existing token aggregation approaches and decouple transformers? A clear structural distinction is needed.

2. Could the authors provide deeper analysis beyond accuracy gains, for example, visual reasoning, interpretability, or sensitivity analysis, to demonstrate real insight?

3. The paper should include experiments based on recent SOTA VLMs, not just out-of-date baselines.

4. Please expand the conclusion to explicitly discuss limitations, the actual scope of applicability, and which scenarios this method should be favored for.

---

> ### Author Response · Authors · 2025-11-20
> **Response to Reviewer FBFx**
>
> We’d like to gently ask Reviewer FBFx to confirm whether the review posted was indeed meant for our paper, or is a review for a different paper that was accidentally posted on for our submission. We raise these suspicions based on the following points:
>
> 1. The summary that our paper "proposes a visual information re-weighting pipeline intended to emphasize key regions in image representations" is inaccurate.
> 2. The terms "Token weighting", "re-weighting", or "token aggregation" are used multiple times in reference to our paper, yet our paper does not focus on these concepts, nor are these terms used in our paper.
> 3. There is a lot of ambiguity in the strengths and weaknesses. Many of the strengths and weaknesses pointed out by the reviewer is not necessarily specific to our paper, but could apply to many other papers related to VLMs/MLLMs.
> 4. Despite the above, the reviewer has submitted the review with very high confidence.
>
> We apologize in advance if our suspicions are incorrect, but we hope the reviewer will give us a confirmation in the near future.
>
> Once we receive confirmation, we will proceed with the rebuttal.
>
> Thank you for your service as an ICLR reviewer.

---

> > ### Comment · Reviewer_FBFx · 2025-11-20
> > **Respond to Authors**
> >
> > Thank you for your prompt response.
> >
> > 1. Based on my current understanding, the proposed strategies, such as Separate QKV Projections and Bidirectional Attention, essentially reorganize the weighting of visual information and thereby influence the interaction patterns among visual features. I interpret this as a form of visual-token re-weighting that differs from the original modality-agnostic mapping and the causal attention mechanism used in standard MLLMs.
> >
> > 2. In addition, the token aggregation process corresponds to the strategy of merging local and global visual features. In this context, I strongly recommend incorporating DeepStack into the discussion, as it is closely related and may provide useful reference points.
> >
> > 3. Not exactly. I also noted that W1 appears to be connected to Reviewer Afru’s W2; W2-3 aligns with Reviewer 8GkR’s W2; and W4-5 correspond to Reviewer Afru’s W1 as well as Reviewer TRSK’s W1, W3, and W4.
> >
> > 4. It is necessary to thoroughly discuss these aspects prior to ICLR submission. In my view, the manuscript has not undergone sufficient background investigation. Without comprehensive research, the contributions have been overstated, which is why I assigned a low score. I suggest revising the paper and fully elaborating on similar, commonly used operations and strategies.
> >
> > 5. If you have any additional questions or concerns, please feel free to raise them, and I will be happy to provide further clarification.
> >
> > Thank you again for your thoughtful response.

---

### Meta-Review · Area_Chair_8Bcm · 2026-01-12

**Summary:**

Across four reviews, the paper is viewed as proposing an interesting set of architectural modifications that improve over LLaVA-style baselines and is generally clearly written, with some evidence from ablations and qualitative analysis. However, reviewers raise substantial concerns about insufficient novelty and weak positioning, including missing key citations and unclear differentiation from closely related token aggregation/decoupled transformer methods and monolithic/encoder-free VLM literature. Multiple reviewers also note that certain components (e.g., hybrid attention masking for visual tokens) are established practices, and the paper does not provide fair comparisons against the most relevant recent alternatives, making the claimed contribution ambiguous.

Equally important, the evaluation is seen as not adequately convincing for acceptance: comparisons are largely confined to the LLaVA family and omit strong modern MLLMs, leaving absolute performance and scalability unclear. Reviewers also highlight missing deployment-focused testing (latency/memory/throughput) and limited robustness/data-efficiency analysis. Given the combination of novelty/related-work gaps, limited baseline coverage, and insufficient evidence of generality beyond a narrow setting, the AC gives a reject decision.

**Reviewer Scores:**

none

---

### Decision · Program_Chairs · 2026-01-26

Reject